# Reservoir-Style Polymeric Drug Delivery Systems: Empirical and Predictive Models for Implant Design

**DOI:** 10.3390/ph15101226

**Published:** 2022-10-03

**Authors:** Linying Li, Chanhwa Lee, Daniela F. Cruz, Sai Archana Krovi, Michael G. Hudgens, Mackenzie L. Cottrell, Leah M. Johnson

**Affiliations:** 1RTI International, 3040 E Cornwallis Road, Research Triangle Park, NC 27709, USA; 2Department of Biostatistics, Gillings School of Global Public Health, University of North Carolina, Chapel Hill, NC 27599, USA; 3Division of Pharmacotherapy and Experimental Therapeutics, Eshelman School of Pharmacy, University of North Carolina, Chapel Hill, NC 27599, USA

**Keywords:** empirical model, implant, long-acting drug delivery system, poly(ε-caprolactone)

## Abstract

Controlled drug delivery systems can provide sustained release profiles, favorable pharmacokinetics, and improved patient adherence. Here, a reservoir-style implant comprising a biodegradable polymer, poly(ε-caprolactone) (PCL), was developed to deliver drugs subcutaneously. This work addresses a key challenge when designing these implantable drug delivery systems, namely the accurate prediction of drug release profiles when using different formulations or form factors of the implant. The ability to model and predict the release behavior of drugs from an implant based on their physicochemical properties enables rational design and optimization without extensive and laborious in vitro testing. By leveraging experimental observations, we propose a mathematical model that predicts the empirical parameters describing the drug diffusion and partitioning processes based on the physicochemical properties of the drug. We demonstrate that the model enables an adequate fit predicting empirical parameters close to experimental values for various drugs. The model was further used to predict the release performance of new drug formulations from the implant, which aligned with experimental results for implants exhibiting zero-order release kinetics. Thus, the proposed empirical models provide useful tools to inform the implant design to achieve a target release profile.

## 1. Introduction

Sustained-release drug delivery systems can provide enhanced therapeutic efficiency with drug levels maintained at a nearly constant rate within a therapeutic window [1,2]. The advantages of these systems often include maximal drug efficacy, minimal side effects, and favorable pharmacokinetics [3]. A common controlled drug delivery system involves a reservoir-based architecture, where a drug core is surrounded by a polymer film. In this system, the drug release rate is controlled by the properties of the polymer such as composition, molecular weight, and film thickness, as well as the physicochemical properties of the enclosed drug, including solubility, drug particle size, and molecular weight [4]. Examples of reservoir-based architectures include injectable microspheres and nanospheres [5,6] with several systems approved by the FDA [7,8,9,10,11,12,13], hydrogel systems [14,15,16,17,18], and implants [19,20,21]. For implants comprising slowly degrading polymers and a constant concentration of drug within the reservoir, the driving force of the drug release is the constant diffusion through the polymer coating, which can result in zero-order release kinetics [4]. In addition, implants offer a promising approach for long-acting drug delivery since they reside under the skin, are discreet to use, and remain reversible during the therapeutic duration if removal is required. One polymer compatible with long-acting implants is semi-crystalline aliphatic polyester poly(ε-caprolactone) (PCL). This polymer is biocompatible, biodegradable by hydrolysis within physiological conditions, and has good mechanical integrity and processability for shaping and manufacture [22]. Due to these characteristics, PCL has been used in various drug delivery systems [23,24,25,26,27] and has shown promise with implants for HIV pre-exposure prophylaxis (HIV PrEP). For instance, Barrett et al. reported a matrix-style implant consisting of an antiretroviral (ARV) drug dispersed within a PCL polymer to deliver 4′-ethynyl-2-fluoro-2′deoxyadenosine (EFdA) [28], an investigational nucleoside reverse transcriptase translocation inhibitor (NRTTI) with subnanomolar antiviral activity, a long half-life, and the potential as a single agent for HIV PrEP [29].

A biodegradable subcutaneous implant with a reservoir architecture was developed by RTI International that comprises a drug formulated with an excipient and encased within a PCL tube for membrane-controlled drug release. Using this architecture, the in vitro and in vivo delivery of tenofovir alafenamide (TAF) for HIV PrEP was demonstrated for multiple months [30,31]. In addition, the long-acting and sustained co-delivery of ARVs and hormones was shown for multipurpose prevention technology (MPT) indications, where progestins and ARVs were simultaneously delivered for HIV PrEP and contraception [32,33]. These studies showed that the release rate of the drug is controlled by modifying the properties of the polymer tubes such as wall thickness, surface area, PCL molecular weight, and polymer composition (e.g., blends of PCL homopolymers of different molecular weight [34]). Specifically, PCL tubes with thin walls, large surface areas, and high molecular weights release drugs faster. The solubility of drugs in the excipient also affects the release rate, where higher rates are typically related to higher solubility of a drug within the excipient.

The use of in vitro experiments can readily inform how implant features affect the drug release rates. Without the aid of in vitro testing, however, predicting the performance of implants with new drug formulations, polymer compositions, or dimensions can prove challenging. A predictive model can inform how the properties of the drug and polymer affect performance and further guide the design of the implant to achieve a target release profile [35,36,37]. Such a predictive model will allow the rapid selection of implant parameters without the need to perform extensive in vitro testing, therefore significantly reducing the time and cost associated with in vitro product development efforts. Empirical models and parameters based on theoretical equations were previously generated for polymeric thin-film matrix and reservoir designs to predict drug release profiles [38]. Herein, we have developed an empirical model to predict drug release rates and profiles for various ARVs for HIV PrEP: the nucleoside reverse transcriptase inhibitors (NRTIs) emtricitabine (FTC), tenofovir alafenamide (TAF_base_), tenofovir alafenamide hemifumarate (TAF_salt_), abacavir (ABC), and lamivudine (3TC); the integrase strand transfer inhibitors (INSTIs) raltegravir potassium (RAL_salt_), dolutegravir sodium (DTG_salt_), and bictegravir (BIC); and the NRTTI EFdA. In addition, the release characteristics of contraceptive hormones including the progestins levonorgestrel (LNG) and etonogestrel (ENG) were evaluated.

This manuscript describes a new linear regression model for predicting the empirical parameters based on the physicochemical properties of seven active pharmaceutical ingredients (APIs) (LNG, ENG, TAF_salt_, TAF_base_, EFdA, BIC, and FTC) based on their molecular weight (MW), solubility in excipient (CsE), solubility in PBS (CsP), pKa, and logP. The empirical model can offer a beneficial approach to inform the product development of implants without requiring excessive experimentation.

## 2. Results and Discussion

### 2.1. In Vitro Performance of PCL Reservoir Implants with Various API Formulations

The implant consists of a biodegradable PCL membrane that encapsulates an API formulated with excipient (Figure 1). To develop empirical and predictive models that describe drug release profiles from this implant, various API formulations were tested in vitro under simulated physiological conditions. Once submerged in the simulated physiological solution, the API that is encapsulated within the implant solubilizes and transports through the PCL membrane via passive diffusion. Because the biodegradation process of PCL is slow and can require several years [39], the faster process of drug delivery is decoupled from biodegradation. When the implant reservoir is saturated, a constant concentration gradient is maintained across the membrane, achieving zero-order release kinetics. A total of seven APIs (BIC, EFdA, ENG, FTC, LNG, TAF_salt_, and TAF_base_; (Table 1)) were each formulated with various pharmaceutical-grade excipients and evaluated using this in vitro method. Appendix A shows exemplary cumulative release profiles of implants with different lengths and wall thicknesses over time. Linear release profiles were achieved for these APIs and the constant release rates were determined.

### 2.2. Predictive Models

As described in the method section, the release of a drug from a reservoir-style implant is driven by a concentration gradient across the polymeric membrane in the following steps: the diffusion of the surrounding aqueous solution into the reservoir of the implant to dissolve the drug, the partitioning of the dissolved drug into the polymer membrane, and finally the diffusion of the dissolved drug through the membrane into the surrounding aqueous media. The diffusion coefficient (*D*) describes the rate of a substance diffusing through the membrane, and the partition coefficient (*k*) determines the ratio of the concentration of a substance for the polymer membrane relative to the aqueous release media. Although *D* and *k* are separate parameters, they are calculated as a single parameter in the empirical model and were not determined independently. Here, we developed a model to predict empirical parameters based on the physicochemical properties of drugs, including MW, solubility in excipient, solubility in PBS, pKa, and logP. The solubility of the drugs within PBS (Table 1) and pharmaceutical-grade excipients (Table 2) were measured by HPLC. As shown in Table 1, the solubility of the APIs in PBS (CsP) varied, with FTC showing the highest solubility and the hormones (ENG, LNG) showing the lowest solubility. Likewise, Table 2 shows that the solubility of the APIs in excipients (CsE) varied across the different combinations. Most APIs tested here were more soluble in excipients such propylene glycol, polysorbate 80, and PEG-based compounds, as compared to excipients such as castor oil and cottonseed oil. Interestingly, LNG showed relatively low solubility in all excipients. Overall, these data were used in next steps to generate the linear regression model.

The fitted linear regression model for predicting Dk in a reservoir system is:(1)$$\log_{10}\left(Dk\right)=-2.052+2.923\times 10^{-3}\cdot \mathrm{MW}-9.067\times 10^{-1}\cdot \log_{10}\left(C_{s}E\right)+2.788\times 10^{-3}\cdot C_{s}P-\;3.196\times 10^{-2}\cdot \mathrm{pKa}+4.872\times 10^{-2}\cdot \mathrm{logP}$$

No property variable was excluded from the model because the regression coefficient Wald test *p*-values were less than 0.05 for all properties. The adjusted $R^{2}$ of the prediction model was 0.75.

Figure 2 compares the predicted $\log_{10}\left(Dk\right)$ values directly to the experimental values. The solid red diagonal line indicates when the prediction and observation are the same, and the blue and orange dashed lines indicate the predictive values within 1 log and 0.5 log, respectively, of the observed values. For all seven APIs, most of the predicted Dk values were in the range of 1/3 to 3 times that of the experimental Dk values. The comparison of predicted $\log_{10}\left(Dk\right)$ values and experimental values for individual drugs are included in Appendix A.

To evaluate the performance of the linear regression model, machine learning prediction models (support vector machine and random forest) were also fit. The mean squared errors of the linear regression, support vector machine, and random forest models were 0.094, 0.124, and 0.089, respectively. The predicted $\log_{10}$(Dk) values from the machine learning models were compared to the experimental values in Appendix A. Overall, all models gave similar results, while the linear regression model provided a simple and explicit prediction equation.

Figure 3 shows representative cumulative release profiles of predicted and experimental fits for BIC, EFdA, ENG, FTC, LNG, and TAF_salt_ from the reservoir-style implants with different configurations. The approximate drug loading for these implants is 116 mg, 22 mg, 7.9 mg, 82mg, 6.9 mg, and 124 mg, respectively. The cumulative release profile represents the average values from three replicate implants and the predicted fits are based on Equation (3) and parameter Dk calculated using Equation (4). The predicted and experimental values for Dk and daily release rates for these implants are tabulated in Appendix A. The cumulative release profiles from the in vitro dissolution assay overlapped with the results from the predictive model, indicating that the model adequately depicts the release profiles from the implants with different configurations.

### 2.3. Utilizing the Predictive Model

We demonstrated that the model provided an adequate fit predicting empirical parameters close to experimental values for the APIs tested above. Here, the developed model was used to predict the release rate of four additional antiretrovirals (RAL_salt_, 3TC, DTG_salt_, and ABC), each having unique physicochemical properties as outlined in Table 3 and Appendix A. Using Equation (4), the predicted Dk values were determined and the resultant comparison of the predicted $\log_{10}$(Dk) values with the experimental values is shown in Figure 4. A good agreement exists between the experimental and predicted Dk values, where all of the predicted values reside between two blue dashed lines, meaning that the predicted Dk values were in the range of 1/10 and 10 times of the experimental Dk values. The mean squared errors of the linear regression, support vector machine, and random forest models were 0.382, 0.498, 0.599, respectively, suggesting an adequate fit of the linear regression model.

Figure 5 illustrates the predicted linear release profiles and experimental data of reservoir-style implants containing an ARV: 3TC, ABC, DTG_salt,_ or RAL_salt_. The approximate drug loading for these implants is 117 mg, 25 mg, 82 mg, and 97 mg, respectively. The proposed predictive models accurately describe the cumulative release profiles from implants containing 3TC and DTG_salt_ formulations, whereas discrepancies exist between experimental data and model predictions for implants containing ABC and RAL_salt_. Since the proposed mathematical model is ideally intended to predict the release profile of implants exhibiting constant zero-order release kinetics, the model does not account for the non-linear release profile that deviates from Fick’s first law of diffusion. As shown in Figure 5, implants containing ABC and RAL_salt_ formulations showed non-linear release profiles due to burst release and/or drug depletion. The ABC formulation demonstrated a pronounced burst release during the first two weeks, which could be attributed to a storage effect, wherein the formulation saturates the entire PCL membrane enclosing the drug reservoir during the storage prior to use. Burst release can occur with reservoir-style implants that encapsulate hydrophilic formulations or have thin polymer walls [30,32]. In addition, the non-linear release profile of the implant with the RAL_salt_ formulation is likely caused by drug depletion. Due to the high release rate of RAL_salt_, the drug concentration in the reservoir quickly falls below the solubility limit, resulting in gradually decreasing release rates over time. Therefore, the applicability of the model is limited when predicting release kinetics that are not zero-order. However, this model closely characterizes the membrane-controlled release process from a reservoir-style implants that reflects zero-order Fickian diffusion, as demonstrated for implants containing 3TC and DTG_salt_.

## 3. Materials and Methods

### 3.1. Solubility and Stability Analysis of the Drug Formulations

FTC, RAL potassium (referred to as RAL_salt_), and DTG sodium (referred to as DTG_salt_) were purchased from Boc Sciences (Shirley, NY, USA). BIC was purchased from AstaTech Inc. (Bristol, PA, USA). 3TC was purchased from Ambeed Inc. (Arlington Heights, IL, USA). ABC was purchased from ThermoFisher Scientific (Bridgewater, NJ, USA). Tenofovir alafenamide (TAF_base_) and tenofovir alafenamide hemifumarate (TAF_salt_) were graciously provided by Gilead Sciences (Foster City, CA, USA). ENG and LNG were procured from AdooQ^®^ Bioscience (Irvine, CA, USA) and Selleck Chemicals LLC (Houston, TX, USA), respectively. EFdA was purchased from Pharmaron (Beijing, China) and Wuxi AppTec (Wuhan, China). To test the solubility and stability of the formulations, the individual drugs were mixed with pharmaceutical-grade, Super Refined^TM^ castor oil (Croda, Cat# SR40890, Snaith, UK), Super Refined^TM^ sesame oil (Croda, Cat# SR40294, Snaith, UK), Crodamol™ ethyl oleate (Croda, Cat# EO-LQ-(MH) ES45252, Snaith, UK), PEG_300_ (Croda, Cat# SR41329, Snaith, UK), PEG_400_ (Croda, Cat# SR40377, Snaith, UK), PEG_600_ (Croda, Cat# SR40269, Snaith, UK), castor oil Etocas 40-SS-(MH) (Croda, Cat# ET48333, Snaith, UK), oleic acid (Croda, Cat# SR40211, Snaith, UK), propylene glycol (Croda, Cat# SR40836, Snaith, UK), or glycerol (Sigma Aldrich, Cat#G6279, St. Louis, MO, USA). The solubility test was conducted by mixing an excess amount of drug with a specific excipient at 37 °C to create a supersaturated solution. The solutions were kept in the incubator at 37 °C for 2 days to determine the solubility of drugs within excipients and for an additional 7 days to assess the solubility and the purity of drugs. After being removed from the incubator, the solutions were then centrifuged while still warm at 1500 rpm for 3 min to separate any undissolved drugs. The supernatants were extracted and analyzed by high-performance liquid chromatography coupled with UV spectroscopy (HPLC-UV) to determine the quantity of the drug dissolved in the excipient. The solubility of the drugs within the excipients was reported as an average solubility measured on day 2 and day 9. The analysis for TAF_salt_ and TAF_base_ was performed using a Waters BEH C18 column (2.1 mm × 50 mm, 1.7 μm) under gradient, reversed-phase conditions with detection at 260 nm. The HPLC analyses of EFdA, ENG, LNG, RAL_salt_, DTG_salt_, and ABC samples were performed using an Agilent Zorbax SB-C8 (4.6 × 150 mm) column on Agilent 1100/1200 HPLC-UV (Agilent Technologies, Santa Clara, CA, USA) with a gradient of 0.01% TFA (solvent A) and acetonitrile (solvent B). BIC samples were analyzed using a Thermo Fisher Hypersil Gold (4.6 × 150 mm) column on an Agilent 1100/1200 HPLC-UV (Agilent Technologies, Santa Clara, CA, USA) with a gradient of 0.01% trifluoroacetic acid (TFA) (solvent A) and acetonitrile (solvent B). 3TC samples were analyzed using an Agilent Zorbax SB-C8 (4.6 × 150 mm) column on Agilent 1100/1200 HPLC-UV (Agilent Technologies, Santa Clara, CA, USA) with a gradient of 0.01% TFA (solvent A) and methanol (solvent B). The saturated solutions were quantitated by linear regression analysis against a 5-point calibration curve.

### 3.2. Implant Fabrication

The research-grade PCL pellets were purchased from Sigma Aldrich (weight average molecular weight (Mw) = 132 kDa, Catalog# 440744, St. Louis, MO, USA). The medical-grade PCL pellets were procured from Corbion (Amsterdam, Netherlands) at different molecular weights: PURASORB PC-12 (Mw = 72 KDa), PURASORB PC-17 (Mw = 106 kDa), and PURASORB PC-41 (Mw = 136 kDa). Medical-grade PCL pellets PC-31 (Mw = 150 kDa) were also procured from Bezwada Biomedical (Hillsborough, NJ, USA). PCL tubes were fabricated via a hot-melt, single-screw extrusion process using solid PCL pellets at GenX Medical (Chattanooga, TN, USA). Before the extrusion process, all the PCL pellets were dried in a compressed air dryer at 60 °C for 4 h. All tubes measured 2.5 mm in outer diameter (OD), as determined using a 3-axis laser measurement system and light microscopy at GenX Medical.

All implants were fabricated in a biosafety cabinet under aseptic conditions using a previously reported method [34]. Prior to starting the in vitro studies, all implants were packed within amber glass vials and sterilized using gamma irradiation (dose range 18–24 kGy) at Steris (Mentor, OH, USA) via continuous exposure to a Cobalt-60 gamma-ray source (Nordion Inc., Ottawa, ON, Canada) for 8 h.

### 3.3. In Vitro Drug Release Studies

For in vitro drug release studies, implants were placed in polypropylene tubes containing 1X phosphate-buffered saline (PBS) (pH 7.4) and incubated at 37 °C within an incubator shaker at 100 rpm. The volume of the buffer and the time intervals for transferring the implants were selected to ensure implants were completely submerged and sink conditions were maintained. The buffer volume in tubes containing implants with LNG, ENG, and RAL_salt_ was 200 mL and the buffer volume in tubes containing implants with TAF, BIC, EFdA, DTG_salt_, 3TC, and ABC was 40 mL. The implants were transferred to fresh PBS buffer twice per week in a biosafety hood under aseptic conditions. During the transfer, a 250 µL aliquot of the release buffer was collected for UV-Vis measurement, whereas a 500 µL aliquot of release media was added to 96-well plates for HPLC analysis. The concentration of the released drug in buffer was determined using either HPLC or UV-Vis. The concentration of BIC and TAF species in the release media were measured by UV-Vis at 260 nm using the Synergy MX multi-mode plate reader (BioTek Instruments, Inc., Winooski, VT, USA). The concentrations of LNG, ENG, EFdA, RAL_salt_, DTG_salt_, 3TC, and ABC were measured with an Agilent 1100/1200 HPLC-UV using an Agilent Zorbax SB-C8 (4.6 × 150 mm) column. The quantity of drug released into the PBS buffer during the time intervals, the cumulative mass of drug released as a function of time, and the daily release rates of drug were calculated as below:

Mass of drug (mg) at a given timepoint (t_n_) = concentration of drug in the release buffer (mg/mL) × volume of release buffer (mL).

Cumulative mass of released drug (mg) at a given timepoint (t_n_) = drug mass at t_0_ + drug mass at t_1_ + … + drug mass at t_n._

Mass of drug released per day (mg/day) = cumulative mass of released drug (mg) at a given timepoint (t_n_)/duration of release (day).

### 3.4. Empirical Models

The release from a reservoir-style implant is predominantly governed by diffusional mass transport through the PCL membrane, which can be described by Fick’s first law of diffusion [57,58]:(2)$$J=-D\frac{d\varphi }{dx}$$
where J is the amount of drug released from the membrane per unit area per unit time (mg/day/mm^2^), D is the diffusion coefficient, *φ* is the concentration, and *x* is the length. When the reservoir is saturated, a constant concentration gradient *dφ*/*dx* is maintained across the membrane, so the rate for drug flux *J* remains constant, achieving zero-order release kinetics. The constant release rate for the membrane-controlled process can be calculated according to the modified diffusion equation [35]:(3)$$J=Dk\frac{C_{s}E}{L}$$
where k is the partition coefficient, $C_{s}E$ is the saturation concentration of the substance within the excipient, and L is the thickness of the PCL membrane. The cumulative mass of drug release $M_{t}\;$at time t can be calculated based on the following equation:(4)$$M_{t}=Dk\frac{C_{s}E}{L}t$$

Based on the experimental data, the empirical parameter Dk can be determined from the slope of a cumulative mass versus time plot using measured design parameters L for each implant and solubility $C_{s}E$ determined by HPLC.

Predictive models were developed using R version 4.2.1 for Dk in the reservoir system as a function of drug properties: molecular weight (MW), solubility in excipient ($C_{s}E$), solubility in PBS ($C_{s}P$), pKa, and logP. Values for $C_{s}E$ and $C_{s}P$ were determined experimentally as reported under Section 2.1, whereas pKa and logP values were obtained through online databanks. Given the skewed distribution of Dk values, linear regression models were fit with $\log_{10}\left(Dk\right)\;$values as the outcome. Transformations (raw, reciprocal, exponential, and logarithm) of property values were considered, and the transformation which had the largest Pearson coefficient with $\log_{10}$(Dk) was chosen as the form of the predictor to be included in the model. Transformed property variables were included in the model only if the corresponding regression coefficient Wald test *p*-value was less than 0.05. Finally, machine learning prediction models (support vector machine and random forest), which are more robust to model misspecification yet do not provide an explicit equation to compute predicted values by hand, were also fit for comparison with the linear regression model.

## 4. Conclusions

Mathematical modeling is a valuable technique for predicting the release profiles of drugs from polymeric implants and for optimizing these implantable systems that exhibit zero-order release kinetics. The empirical models presented here offer a systematic approach to determine the empirical parameters that define the membrane-controlled drug release profiles from implants with a reservoir configuration. The empirical parameter, Dk, is correlated to the key physicochemical properties of the drug (MW, solubility in excipient and in PBS, pKa, and logP) and once determined can enable the prediction of well-suited drug formulations to achieve a target release rate for a particular medical indication. The utility of the empirical model is further exemplified when designing implants containing several new ARVs for HIV treatment. In addition, we detailed the applicability for using the predictive model. Overall, the empirical model provides useful tools to guide the implant design, and the approach for developing predictive models could be extended to other drug-eluting polymeric implant systems.

## Figures and Tables

**Figure 1 pharmaceuticals-15-01226-f001:**
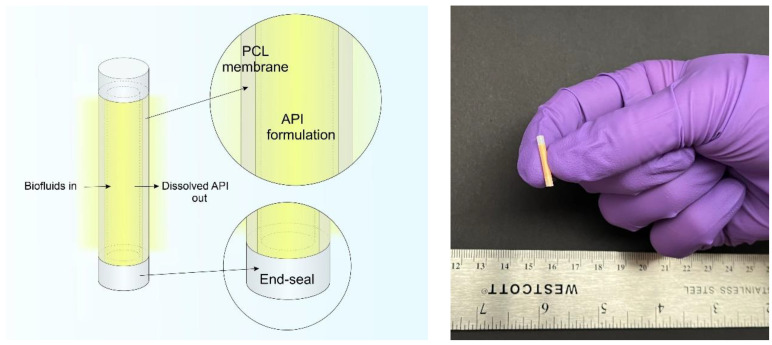
(**Left**) A schematic of a PCL reservoir-style implant containing formulated API inside the reservoir and the dissolved drug releasing into to the surrounding medium. (**Right**) A digital camera image of a 10mm-long biodegradable implant containing surrogate drug formulation.

**Figure 2 pharmaceuticals-15-01226-f002:**
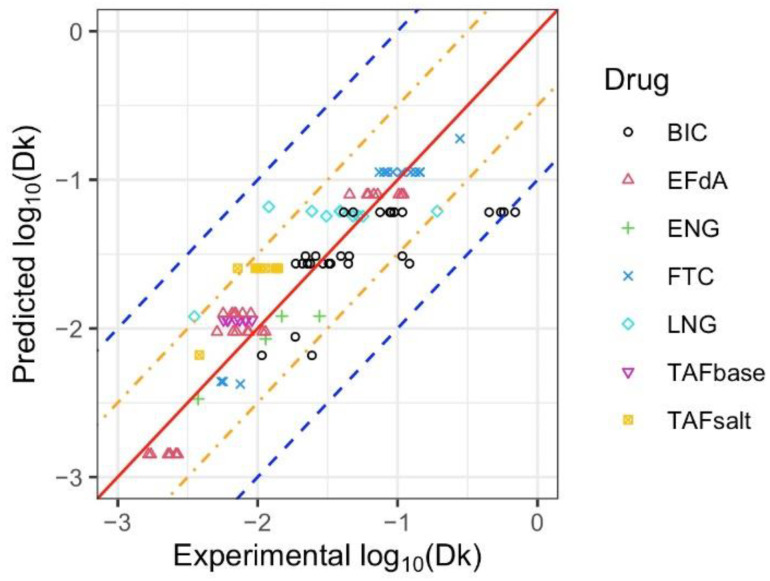
Predicted and experimental values of $\log_{10}\left(Dk\right)\;$for BIC, EFdA, ENG, FTC, LNG, TAF_base_, and TAF_salt_. The solid red diagonal line indicates when the prediction and observation are the same, the orange and blue dashed lines indicate predictive values within 0.5 log and 1 log of the observed values, respectively. Different symbols represent each drug, with multiple points for the same symbol representing different configurations of drug formulation or implant design tested for each drug.

**Figure 3 pharmaceuticals-15-01226-f003:**
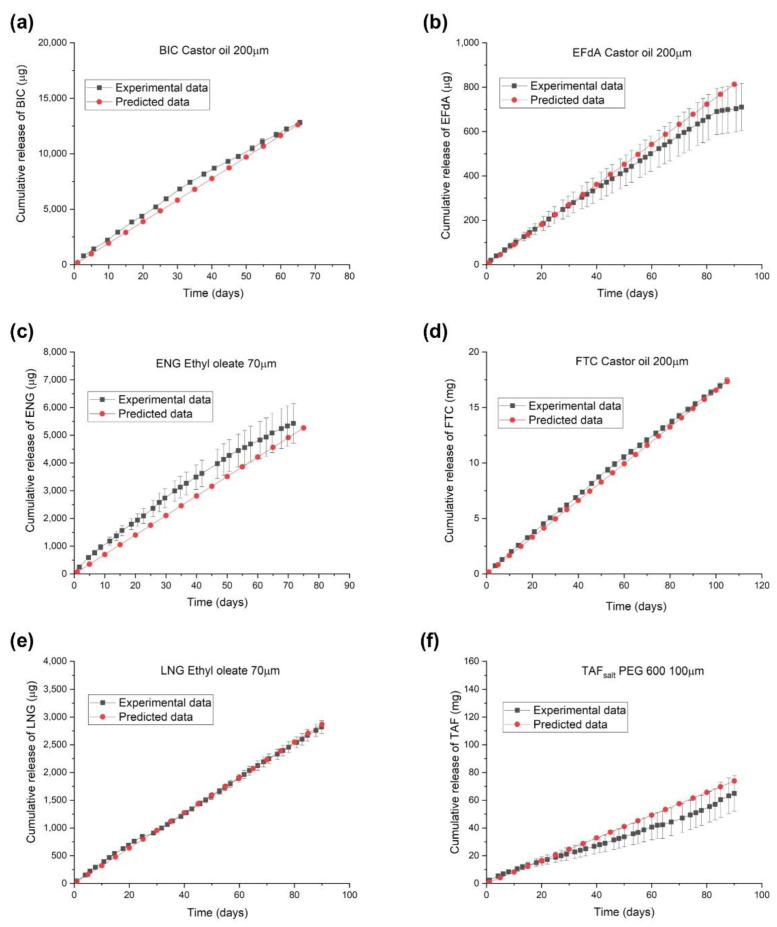
Predicted and experimental release profiles of implants containing a single drug with individual configurations: (**a**) BIC (castor oil, PC-17, 40 mm length, 200 μm wall thickness, drug to excipient ratio: 2:1), (**b**) EFdA (castor oil, PC-17, 10 mm length, 200 μm wall thickness, drug to excipient ratio: 1:1), (**c**) ENG (ethyl oleate, Sigma, 10 mm length, 70 μm wall thickness, drug to excipient ratio: 1:4), (**d**) FTC (castor oil, PC-17, 40 mm length, 200 μm wall thickness, drug to excipient ratio: 1:1), (**e**) LNG (ethyl oleate, Sigma, 10 mm length, 70 μm wall thickness, drug to excipient ratio: 1:4), and (**f**) TAF_salt_ (PEG_600_, Sigma, 40 mm length, 100 μm wall thickness, drug to excipient ratio: 2:1).

**Figure 4 pharmaceuticals-15-01226-f004:**
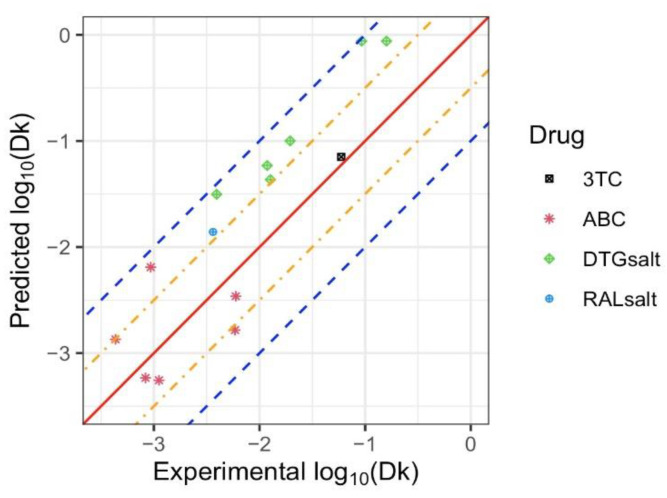
Predicted and experimental values of $\log_{10}\left(Dk\right)\;$. for 3TC, ABC, DTG_salt_, and RAL_salt_. The solid red diagonal line indicates when the prediction and observation are the same, the orange and blue dashed lines indicate predictive values within 0.5 log and 1 log of the observed values, respectively. Different symbols represent each drug, with multiple points for the same symbol representing different implant configurations tested for each drug.

**Figure 5 pharmaceuticals-15-01226-f005:**
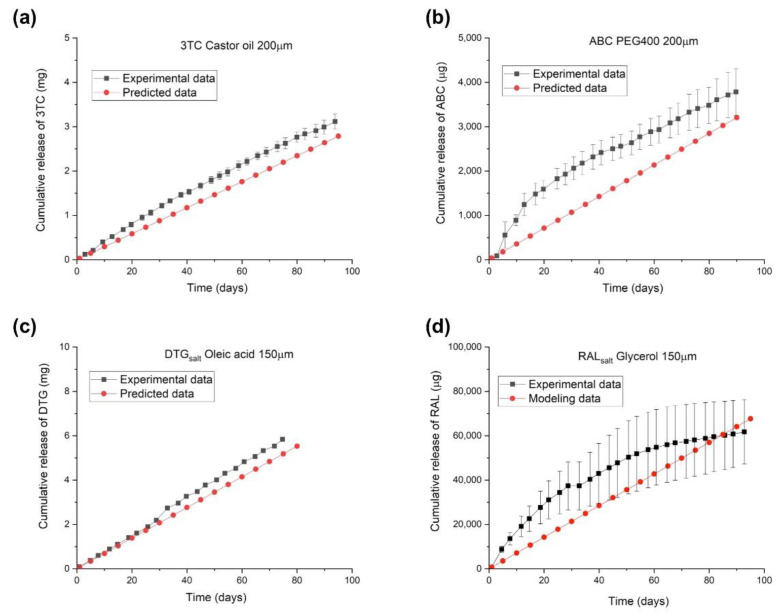
Predicted and experimental release profiles of implants containing a single ARV with individual configurations: (**a**) 3TC (castor oil, PC-17, 40 mm length, 200 μm wall thickness, drug to excipient ratio: 2:1), (**b**) ABC (PEG_400_, PC-17, 10 mm length, 200 μm wall thickness, drug to excipient ratio: 2:1), (**c**) DTG_salt_ (oleic acid, PC-31, 10 mm length, 150 μm wall thickness, drug to excipient ratio: 2:1), and (**d**) RAL_salt_ (glycerol, PC-31, 40 mm length, 150 μm wall thickness, drug to excipient ratio: 2:1).

**Table 1 pharmaceuticals-15-01226-t001:** Physical properties and sources of APIs used to develop the model.

API	Suppliers	logP	Molecular Weight (MW) (Da)	Solubility in PBS (CsP) (mg/mL)	pKa
LNG	Selleck	3.8 [40]	312.5	0.0071	17.9 [41]
ENG	AdooQ	3.3 [42]	324.6	0.0093	10.4 [43]
TAF_salt_	Gilead	1.49 [44]	534.5	11.59	11.36 [44]
TAF_base_	Gilead	1.8 [45]	476.5	4.98	3.96 [46]
EFdA	Wuxi/Pharm	−1.19 [47]	293.2	1.05	13.32 [48]
BIC	AstaTech	1.71 [49]	449.4	0.17	9.81 [50]
FTC	BOC Sciences	−0.43 [51]	247.2	165.6	2.65 [51]

**Table 2 pharmaceuticals-15-01226-t002:** The solubility of the API within various pharmaceutical-grade excipients.

Excipient	BIC Solubility (mg/mL)	Wuxi EFdA Solubility (mg/mL)	Pharm EFdA Solubility (mg/mL)	ENG Solubility (mg/mL)	FTC Solubility (mg/mL)	LNG solubility (mg/mL)	TAF_base_ Solubility (mg/mL)	TAF_salt_ Solubility (mg/mL)
Castor Oil	4.35 ± 1.58	1.81 ± 0.10	2.50 ± 0.18	16.20 ± 0.76	0.906 ± 0.14	1.24 ± 0.21	16.75 ± 0.23	12.4 ± 0.01
Cottonseed Oil	2.19 ± 1.13	0.04 ± 0.01	0.057 ± 0.002	3.98 ± 0.07	0.011 ± 0.002	0.51 ± 0.07	0.19 ± 0.18	0.168 ± 0.004
Ethyl Oleate	0.62 ± 0.24	0.04 ± 0.01	0.05 ± 0.001	5.60 ± 0.05	0.015 ± 0.002	0.59 ± 0.03	0.21 ± 0.15	0.11 ± 0.01
Glycerol	3.81 ± 1.21	21.9 ± 0.22	11.5 ± 0.14	2.02 ± 1.20	36.9 ± 1.69	0.55 ± 0.23	29.19 ± 2.73	41.8 ± 0.55
Oleic Acid	15.9 ± 0.29	0.71 ± 0.23	0.054 ± 0.001	4.73 ± 0.37	0.5 ± 0.01	0.50 ± 0.14	52.96 ± 2.08	59.9 ± 0.71
PEG_300_	24.8 ± 1.42	69.89 ± 0.86	11.4 ± 0.16	32.95 ± 0.93	37.0 ± 3.71	3.68 ± 0.33	66.93 ± 3.79	65.2 ± 0.36
PEG_400_	24.8 ± 5.83	68.37 ± 3.52	14.2 ± 0.13	32.82 ± 1.13	37.2 ± 1.76	3.83 ± 0.23	67.05 ± 2.96	39.9 ± 0.18
PEG_600_	25.4 ± 1.83	62.87 ± 0.52	14.2 ± 0.13	31.10 ± 1.26	39.0 ± 0.82	3.81 ± 0.07	59.60 ± 3.22	57.6 ± 0.44
PEG_40_ Castor Oil	22.1 ± 3.00	37.54 ± 0.60	22.2 ± 0.21	28.02 ± 1.71	21.2 ± 0.21	4.13 ± 0.46	18.37 ± 1.46	28.4 ± 0.22
Polysorbate 80	24.1 ± 0.72	35.02 ± 1.00	16.9 ± 0.047	25.44 ± 1.55	14.8 ± 0.17	3.49± 0.54	19.06 ± 3.12	28.5 ± 0.85
Propylene Glycol	24.2 ± 4.09	41.45 ± 1.22	16.9 ± 0.05	18.66 ± 1.28	38.6 ± 0.49	3.49 ± 0.54	63.59 ± 4.15	75.8 ± 0.86
Sesame Oil	1.75 ± 1.02	0.03 ± 0.01	0.22 ± 0.01	3.74 ± 0.06	0.020 ± 0.006	0.54 ± 0.04	0.06 ± 0.004	0.34 ± 0.11

**Table 3 pharmaceuticals-15-01226-t003:** Physical properties and sources of APIs for model validation.

API	Suppliers	LogP	Molecular Weight (MW) (Da)	$\mathrm{Solubility}\;\mathrm{in}\;\mathrm{PBS}\;(C_{s}P)\;(\mathrm{mg}/\mathrm{mL})$	pKa
3TC	Ambeed	−1.4 [52]	229.3	78.1	14.29 [52]
ABC	TCI America	1.335 [53]	286.3	2.55	4.8 [54]
DTG_salt_	BOC Sciences	2.2 [55]	441.4	0.08	8.2 [56]
RAL_salt_	BOC Sciences	1.59 [56]	482.5	64.99	7.02 [56]

## Data Availability

Data is contained within the article and Appendix A.

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
