# Peer review of "Reservoir-Style Polymeric Drug Delivery Systems: Empirical and Predictive Models for Implant Design"

_pharmaceuticals, 2022, doi:10.3390/ph15101226_

Round 1

Reviewer 1 Report

The manuscript concerns the implication of empirical model to evaluate the drug release from subcutaneous implants in experiments in vitro. The empirical models based on the established properties of the well-knonw drugs may be of interest to enhcnase the modern testing protocols in the development of novel drug formulations. I believe, experimental studies can not be completely replaced by empirical models, however, can be strongly supported by them. The study is well designed, the methods are adequate, and the conclusions are well supported by the reported results. The manuscript is suitable for publication in Pharmaceuticals after very minor revision. 

In the experimental section, I would like the authors to concretize how the cumulative release was calculated in experiments in vitro, and how the concentration of the released drugs was measured in the PBS?
